# Non-Linear Domain Adaptation with Boosting

**Carlos Becker**[*]          **C. Mario Christoudias**          **Pascal Fua**

CVLab, École Polytechnique Fédérale de Lausanne, Switzerland

`firstname.lastname@epfl.ch`

## Abstract

A common assumption in machine vision is that the training and test samples are drawn from the same distribution. However, there are many problems when this assumption is grossly violated, as in bio-medical applications where different acquisitions can generate drastic variations in the appearance of the data due to changing experimental conditions. This problem is accentuated with 3D data, for which annotation is very time-consuming, limiting the amount of data that can be labeled in new acquisitions for training. In this paper we present a multi-task learning algorithm for domain adaptation based on boosting. Unlike previous approaches that learn task-specific decision boundaries, our method learns a single decision boundary in a shared feature space, common to all tasks. We use the *boosting-trick* to learn a non-linear mapping of the observations in each task, with no need for specific a-priori knowledge of its global analytical form. This yields a more parameter-free domain adaptation approach that successfully leverages learning on new tasks where labeled data is scarce. We evaluate our approach on two challenging bio-medical datasets and achieve a significant improvement over the state of the art.

## 1   Introduction

Object detection and segmentation approaches often assume that the training and test samples are drawn from the same distribution. There are many problems in Computer Vision, however, where this assumption can be grossly violated, such as in bio-medical applications that usually involve expensive and complicated data acquisition processes that are not easily repeatable. As illustrated in Fig. 1, this can result in newly-acquired data that is significantly different from the data used for training. As a result, a classifier trained on data from one acquisition often cannot generalize well to data obtained from a new one. Furthermore, although it is possible to expect supervised data from a new acquisition, it is unreasonable to expect the practitioner to re-label large amounts of data for each new image that is acquired, particularly in the case of 3D image stacks.

A possible solution is to treat each acquisition as a separate, but related classification problem, and exploit their possible relationship to learn from the supervised data available across all of them. Typically, each such classification problem is called a *task*, which is associated with a *domain*. For example, for Fig. 1(a,b) the task is mitochondria segmentation in both acquisitions. However, the domains are different, namely Striatum and Hippocampus EM stacks. Techniques in *domain adaptation* [1] and more generally *multi-task learning* [2, 3] seek to leverage data from a set of different yet related tasks or domains to help learn a classifier in a seemingly new task. In domain adaptation, it is typically assumed that there is a fairly large amount of labeled data in one domain, commonly referred to as the *source domain*, and that a limited amount of supervision is available in the other, often called the *target domain*. Our goal is to exploit the labeled data in the source domain to learn an accurate classifier in the target domain despite having only a few labeled samples in the latter.

---
[*]This work was supported in part by the ERC grant MicroNano.

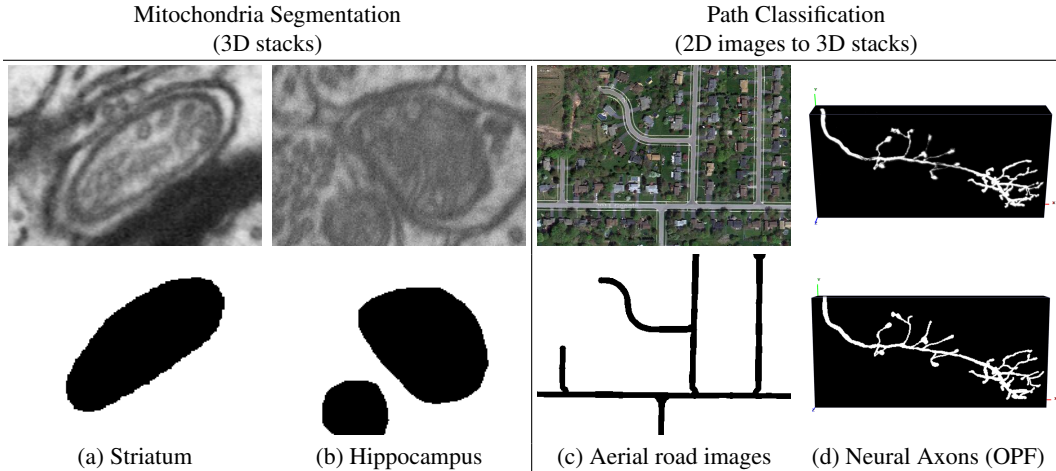

| Mitochondria Segmentation | Path Classification |
|---|---|
| (3D stacks) | (2D images to 3D stacks) |

(a) Striatum      (b) Hippocampus      (c) Aerial road images      (d) Neural Axons (OPF)

Figure 1: (a,b) Slice cuts from two 3D Electron Microscopy acquisitions from different parts of the brain of a rat. (c,d) 2D aerial road images and 3D neural axons from Olfactory Projection Fibers (OPF). Top and bottom rows show example images and ground truth respectively.

The data acquisition problem is unique to many multi-task learning problems, however, in that each task is in fact the same, but what has changed is that the features across different acquisitions have undergone some unknown transformation. That is to say that each task can be well described by a single decision boundary in some common feature space that preserves the task-relevant features and discards the domain specific ones corresponding to unwanted acquisition artifacts. This contrasts the more general multi-task setting where each task is comprised of both a common and task-specific boundary, even when mapped to a common feature space, as illustrated in Fig. 2. A method that can jointly optimize over the common decision boundary and shared feature space is therefore desired.

Linear latent variable methods such as those based on Canonical Correlation Analysis (CCA) [4, 5] can be applied to learn a shared feature space across the different acquisitions. However, the situation is further complicated by the fact that the unknown transformations are generally non-linear. Although kernel methods can be applied to model the non-linearity [4, 6, 7], this requires the existence of a well-defined kernel function that can often be difficult to specify a priori. Also, the computational complexity of kernel methods scales quadratically with the number of training examples, limiting their application to large datasets.

In this paper we propose a solution to the data acquisition problem and devise a method that can jointly solve for the non-linear decision boundary and transformations across tasks. As illustrated in Fig. 2 our approach maps features from possibly high-dimensional, task-specific feature spaces to a low-dimensional space common to all tasks. We assume that only the mappings are task-dependent and that in the shared space the problem is linearly separable and the decision boundary is common to all tasks. We use the *boosting-trick* [8, 9, 10] to simultaneously learn the non-linear task-specific mappings as well as the decision boundary, with no need for specific a-priori knowledge of their global analytical form. This yields a more parameter-free domain adaptation approach that successfully leverages learning on new tasks where labeled data is scarce.

We evaluate our approach on the two challenging bio-medical datasets depicted by Fig. 1. We first consider the classification of curvilinear structures in 3D image stacks of Olfactory Projection Fibers (OPF) [11] using labeled 2D aerial road images. We then perform mitochondria segmentation in large 3D Electron Microscopy (EM) stacks of neural rat tissue, demonstrating the ability of our algorithm to leverage labeled data from different data acquisitions on this challenging task. On both datasets our approach obtains a significant improvement over using labeled data from either domain alone and outperforms recent multi-task learning baseline methods.

## 2 Related Work

Initial ideas to multi-task learning exploited supervised data from related tasks to define a form of regularization in the target problem [2, 12]. In this setting, related tasks, also sometimes referred to

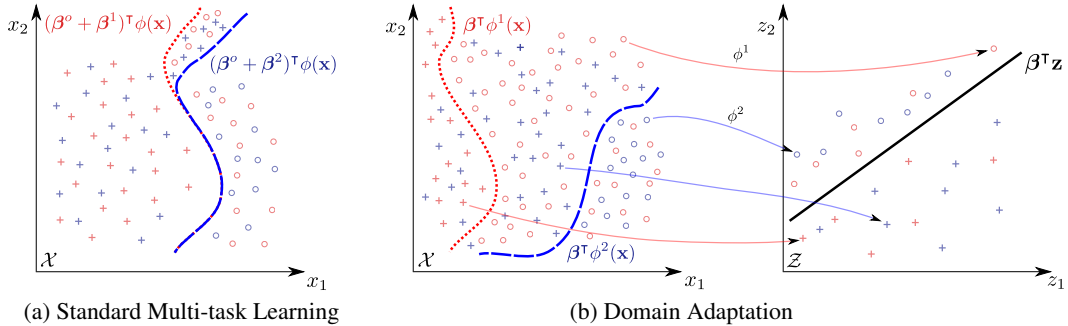

<center>(a) Standard Multi-task Learning        (b) Domain Adaptation</center>

Figure 2: Illustration of the difference between (a) standard Multi-task Learning (MTL) and (b) our Domain Adaptation (DA) approach on two tasks. MTL assumes a single, pre-defined transformation $\phi(\mathbf{x}) : \mathcal{X} \to \mathcal{Z}$ and learns shared and task-specific linear boundaries in $\mathcal{Z}$, namely $\boldsymbol{\beta}^o$, $\boldsymbol{\beta}^1$ and $\boldsymbol{\beta}^2 \in \mathcal{Z}$. In contrast, our DA approach learns a single linear boundary $\boldsymbol{\beta}$ in a common feature space $\mathcal{Z}$, and task-specific mappings $\phi^1(\mathbf{x})$, $\phi^2(\mathbf{x}) : \mathcal{X} \to \mathcal{Z}$. Best viewed in color.

as *auxiliary problems* [13], are used to learn a latent representation and find discriminative features shared across tasks. This representation is then *transferred* to the target task to help regularize the solution and learn from fewer labeled examples. The success of these approaches crucially hinges on the ability to define auxiliary tasks. Although this can be easily done in certain situations, e.g., as in [13], in many cases it is unclear how to generate them and the solution can be limiting, especially when provided only a few auxiliary problems. Unlike such methods, our approach is able to find an informative shared representation even with as little as one related task.

More recent multi-task learning methods jointly optimize over both the shared and task-specific components of each task [3, 14, 10, 15]. In [3] it was shown how the two step iterative optimization of [13] can be cast into a single convex optimization problem. In particular, for each task their approach computes a linear decision boundary defined as a linear combination between a shared hyperplane, shared across tasks, and a task-specific one in either the original or a kernelized feature space. This idea was later further generalized to allow for more generic forms [14, 16, 17, 15], as in [14] that investigated the use of a hierarchically combined decision boundary. The use of boosting for multi-task learning was explored in [10] as an alternative to kernel-based approaches. For each task they optimize for a shared and task-specific decision boundary similar to [3], except non-linearities are modeled using a boosted feature space. As with other methods, however, additional parameters are required to control the degree of sharing between tasks that can be difficult to set, especially when one or more tasks have only a few labeled samples.

For many problems, such as those common to domain adaptation [1], the decision problem is in fact the same across tasks, however, the features of each task have undergone some unknown transformation. Feature-based approaches seek to uncover this transformation by learning a mapping between the features across tasks [18, 19, 7]. A cross-domain Mahalanobis distance metric was introduced in [18] that leverages across-task correspondences to learn a transformation from the source to target domain. A similar method was later developed in [20] to handle cross-domain feature spaces of a different dimensionality. Shared latent variable models have also been proposed to learn a shared representation across multiple feature sources or tasks [4, 19, 6, 7, 21].

Feature-based methods generally rely on the kernel-trick to model non-linearities that requires the selection of a pre-defined kernel function and is difficult to scale to large datasets. In this paper, we exploit the *boosting-trick* [10] to handle non-linearities and learn a shared representation across tasks, overcoming these limitations. This results in a more parameter-free, scalable domain adaptation approach that can leverage learning on new tasks where labeled data is scarce.

## 3 Our Approach

We consider the problem of learning a binary decision function from supervised data collected across multiple tasks or domains. In our setting, each task is an instance of the same underlying decision problem, however, its features are assumed to have undergone some unknown non-linear transformation.

Assume that we are given training samples $X^t = \{\mathbf{x}_i^t, y_i^t\}_{i=1}^{N^t}$ from $t = 1, \ldots, T$ tasks, where $\mathbf{x}_i^t \in \mathbb{R}^D$ represents a feature vector for sample $i$ in task $t$ and $y_i^t \in \{-1, 1\}$ its label. For each task, we seek to learn a non-linear transformation, $\phi^t(\mathbf{x}^t)$, that maps $\mathbf{x}^t$ to a common, task-independent feature space, $\mathcal{Z}$, accounting for any unwanted feature shift. Instead of relying on cleverly chosen kernel functions we model each transformation using a set of task-specific non-linear functions $\mathcal{H}^t = \{h_1^t, \ldots, h_M^t\}, h_j^t : \mathbb{R}^D \to \mathbb{R}$, to define $\phi^t : \mathcal{X}^t \to \mathcal{Z}$ as $\phi^t(\mathbf{x}^t) = [h_1^t(\mathbf{x}^t), \ldots, h_M^t(\mathbf{x}^t)]^\intercal$.

A wide variety of task-specific feature functions can be explored within our framework. We consider functions of the form,

$$h_j^t(\mathbf{x}^t) = h_j(\mathbf{x}^t - \boldsymbol{\tau}_j^t), \quad j = 1, \ldots, M \tag{1}$$

where $\mathcal{H} = \{h_1, \ldots, h_M\}$ are shared across tasks and $\boldsymbol{\tau}_j^t \in \mathbb{R}^D$. This seems like an appropriate model in the case of feature shift between tasks, for example due to different acquisition parameters. Each $h_j$ can be interpreted as a weak non-linear predictor of the task label and in practice $M$ is large, possibly infinite. In what follows, we set $\mathcal{H}$ to be the set of regression trees or stumps [8] that in combination with $\boldsymbol{\tau}^t$ can be used to model highly complex, non-linear transformations.

Assuming that the problem is linearly separable in $\mathcal{Z}$ the predictive function $f_t(\cdot) : \mathbb{R}^D \to \mathbb{R}$ for each task can then be written as

$$f_t(\mathbf{x}) = \boldsymbol{\beta}^\intercal \phi^t(\mathbf{x}^t) = \sum_{j=1}^M \beta_j h_j(\mathbf{x}^t - \boldsymbol{\tau}_j^t) \tag{2}$$

where $\boldsymbol{\beta} \in \mathbb{R}^M$ is a linear decision boundary in $\mathcal{Z}$ that is common to all tasks. This contrasts previous approaches to multi-task learning such as [3, 10] that learn a separate decision boundary per task and, as we show later, is better suited for problems in domain adaptation.

We learn the functions $f_t(\cdot)$ by minimizing the exponential loss on the training data across each task

$$\boldsymbol{\beta}^*, \Gamma^* = \min_{\boldsymbol{\beta}, \Gamma} \sum_{t=1}^T L(\boldsymbol{\beta}, \Gamma^t; X^t), \tag{3}$$

where

$$L(\boldsymbol{\beta}, \Gamma^t; X^t) = \sum_{i=1}^{N^t} \exp\left[-y_i^t f_t(\mathbf{x}_i^t)\right] = \sum_{i=1}^{N^t} \exp\left[-y_i^t \sum_{j=1}^M \beta_j h_j(\mathbf{x}_i^t - \boldsymbol{\tau}_j^t)\right], \tag{4}$$

and $\Gamma = [\Gamma^1, \ldots, \Gamma^T]$ with $\Gamma^t = [\boldsymbol{\tau}_1^t, \ldots, \boldsymbol{\tau}_M^t]$.

The explicit minimization of Eq. (3) can be very difficult, since in practice, $M$ can be prohibitively large and the $h_j$'s are typically discontinuous and highly non-linear. Luckily, this is a problem for which boosting is particularly well suited [8], as it has been demonstrated to be an effective method for constructing a highly accurate classifier from a possibly large collection of weak prediction functions. Similar to the kernel-trick, the resulting *boosting-trick* [8, 9, 10] can be used to define a non-linear mapping to a high dimensional feature space for which we assume the data to be linearly separable. Unlike the kernel-trick, however, the boosting-trick defines an explicit mapping for which $\boldsymbol{\beta}$ is assumed to be sparse [22, 10].

We propose to use gradient boosting [8, 9] to solve for $f_t(\cdot)$. Given any twice-differentiable loss function, gradient boosting minimizes the loss in a stage-wise manner for iterations $k = 1$ to $K$. In particular, we use the quadratic approximation introduced by [9]. When applied to minimize Eq. (3), the goal at each boosting iteration is to find the weak learner $\tilde{h} \in \mathcal{H}$ and the set of $\{\tilde{\boldsymbol{\tau}}^1, \ldots, \tilde{\boldsymbol{\tau}}^T\}$ that minimize

$$\sum_{t=1}^T \left( \sum_{i=1}^{N^t} w_{ik}^t \left[ \tilde{h}(\mathbf{x}^t - \tilde{\boldsymbol{\tau}}^t) - r_{ik}^t \right]^2 \right), \tag{5}$$

where $w_{ik}^t$ and $r_{ik}^t$ can be computed by differentiating the loss of Eq. (4), obtaining $w_{ik}^t = e^{-y_i^t f_t(\mathbf{x}_i^t)}$ and $r_{ik}^t = y_i^t$. Once $\tilde{h}$ and $\{\tilde{\boldsymbol{\tau}}^1, \ldots, \tilde{\boldsymbol{\tau}}^T\}$ are found, a line-search procedure is applied to determine

**Algorithm 1** Non-Linear Domain Adaptation with Boosting

---

**Input:** Training samples and labels for $T$ tasks $X^t = \{(\mathbf{x}_i^t, y_i^t)\}_{i=1}^{N^t}$
Number of iterations $K$, shrinkage factor $0 < \gamma \leq 1$

1: Set $f_t(\cdot) = 0 \ \forall \ t = 1, \ldots, T$

2: **for** $k = 1$ to $K$ **do**

3:    Let $w_{ik}^t = e^{-y_i^t f_t(\mathbf{x}_i^t)}$ and $r_{ik}^t = y_i^t$

4:    Find $\left\{ \tilde{h}(\cdot), \tilde{\boldsymbol{\tau}}^1, \ldots, \tilde{\boldsymbol{\tau}}^T \right\} = \underset{h \in \mathcal{H}, \boldsymbol{\tau}^1, \ldots, \boldsymbol{\tau}^T}{\operatorname{argmin}} \sum_{t=1}^{T} \sum_{i=1}^{N^t} w_{ik}^t \left[ h(\mathbf{x}_i^t - \boldsymbol{\tau}^t) - r_{ik}^t \right]^2$

5:    Find $\tilde{\alpha}$ through line search: $\tilde{\alpha} = \underset{\alpha}{\operatorname{argmin}} \sum_{t=1}^{T} \sum_{i=1}^{N^t} \exp\left[ -y_i^t \left( f_t(\mathbf{x}_i^t) + \alpha \, \tilde{h}(\mathbf{x}_i^t - \tilde{\boldsymbol{\tau}}^t) \right) \right]$

6:    Set $\tilde{\beta} = \gamma \, \tilde{\alpha}$

7:    Update $f_t(\cdot) = f_t(\cdot) + \tilde{\beta} \, \tilde{h}(\cdot - \tilde{\boldsymbol{\tau}}^t) \ \ \forall \ t = 1, \ldots, T$

8: **end for**

9: **return** $f_t(\cdot) \ \ \forall \ t = 1, \ldots, T$

---

the optimal weighting for $\tilde{h}$ and the predictive functions $f_t(\cdot)$ are updated, as described in Alg. 1. Shrinkage may be applied to help regularize the solution, particularly when using powerful weak learners such as regression trees [8].

Our proposed approach is summarized in Alg. 1. The main difficulty in applying this method is in line 4, which finds the optimal values of $\tilde{h}$ and $\{\tilde{\boldsymbol{\tau}}^1, \ldots, \tilde{\boldsymbol{\tau}}^T\}$ that minimize Eq. 5. This can be very expensive, depending on the type of weak learners employed. In the next section we show that regression trees and boosted stumps can be used efficiently to minimize Eq. (5) at train time.

### 3.1 Weak Learners

Regression trees have proven very effective when used as weak learners with gradient boosting [23]. An important advantage is that training regression trees needs practically no parameter tuning and is very efficient when a greedy top-down approach is used [8].

Decision stumps represent a special case of single-level regression trees. Despite their simplicity, they have been demonstrated to achieve a high performance in challenging tasks such as face and object detection [24, 25]. In cases where feature dimensionality $D$ is very large, decision stumps may be preferred over regression trees to reduce training time.

**Regression Trees:** We use trees whose splits operate on a single dimension of the feature vector, and follow the top-down greedy tree learning approach described in [8]. The top split is learned first, seeking to minimize

$$\underset{\substack{n \in \{1, \ldots, D\}, \\ \eta_1, \eta_2, \{\tau^1, \ldots, \tau^T\}}}{\operatorname{argmin}} \sum_{t=1}^{T} \left( \sum_{i=1}^{N^t} \mathbf{1}_{\{\mathbf{x}_i^t[n] - \tau^t\}} w_{ik}^t \left[ \eta_1 - r_{ik}^t \right]^2 + \sum_{i=1}^{N^t} \bar{\mathbf{1}}_{\{\mathbf{x}_i^t[n] - \tau^t\}} w_{ik}^t \left[ \eta_2 - r_{ik}^t \right]^2 \right), \quad (6)$$

where $\mathbf{x}[n] \in \mathbb{R}$ denotes the value of the $n^{\text{th}}$ dimension of $\mathbf{x}$, $\mathbf{1}_{\{\cdot\}}$ is the indicator function, and $\bar{\mathbf{1}}_{\{\cdot\}} = 1 - \mathbf{1}_{\{\cdot\}}$. The difference w.r.t. classic regression trees is that, besides learning the values of $\eta_1$, $\eta_2$ and $n$, our approach requires the tree to also learn a threshold $\tau^t \in \mathbb{R}$ per task. Given that each split operates on a single attribute $\mathbf{x}[n]$, the resulting $\tilde{\boldsymbol{\tau}}^t$ is sparse, and learned one component at a time as the tree is built.

Once the top split is learned, a new split is trained on each of its child leaves, in a recursive manner. This process is repeated until the maximum depth $L$, given as a parameter, is reached, or there are not enough samples to learn a new node at a given leaf.

**Decision Stumps:** Decision stumps consist of a single split and return values $\eta_1, \eta_2 = \pm 1$. If also $r_{ik}^t = \pm 1$, which is true when boosting with the exponential loss, then it can be demonstrated that minimizing Eq (6) can be separated into $T$ independent minimization problems for all $D$ attributes for each $n$. Once this is done, a quick search can be performed to determine the $n$ that minimizes Eq. (6). This makes decision stumps feasible for large-scale applications with very high dimensional feature spaces.

In the special case of the exponential loss and decision stumps, it can be shown that Alg. 1 reduces to a procedure similar to classic AdaBoost [26], with the exception that weak learner search is done in the multi-task manner described above.

## 4  Evaluation

We evaluated our approach on two challenging domain adaptation problems for which annotation is very time-consuming, representative of the problems we seek to address. We first describe the datasets, our experimental setup and baselines, and finally present and discuss the obtained results.

### 4.1  Datasets

**Path Classification**  Tracing arbors of curvilinear structures is a well studied problem that finds applications in a broad range of fields from neuroscience to photogrammetry. We consider the detection of 3D curvilinear structures in 3D image stacks of Olfactory Projection Fibers (OPF) using 2D aerial road images (see Fig. 1(c,d)). For this problem, the task is to predict whether a tubular path between two image locations belongs to a curvilinear structure. We used a publicly-available dataset [11] of 2D aerial images of road networks as the source domain and 3D stacks of Olfactory Projection Fibers (OPF) from the DIADEM challenge as the target domain. The source domain consists of six fully-labeled 2D aerial road images and the target domain contains eight fully-labeled 3D stacks. We aim at using large amounts of labeled data from 2D road images to leverage learning in the 3D stacks. This is a clear scenario where transfer learning can be highly beneficial, because labeling 2D images is much easier than annotating 3D stacks. Therefore, being able to take advantage of 2D data is essential to reduce tedious 3D labeling effort.

**Mitochondria Segmentation:**  Mitochondria are organelles that play an important role in cellular functioning. The goal of this task is to segment mitochondria from large 3D Electron Microscopy (EM) stacks of 5 nm voxel size, acquired from the brain of a rat. As in the path classification problem, 3D annotations are time-consuming and exploiting already-annotated stacks is essential to speed up analysis. The source domain is a fully-labeled EM stack from the Striatum region of 853x506x496 voxels with 39 labeled mitochondria. The target domain consists of two stacks acquired from the Hippocampus, one a training volume of size 1024x653x165 voxels and the other a test volume that is 1024x883x165 voxels, with 10 and 42 labeled mitochondria in each respectively. The target test volume is fully-labeled, while the training one is partially annotated, similar to a real scenario. To capture contextual information, state-of-the-art methods typically use filter response vectors of more than 100k dimensions, which in combination with the size of this dataset, makes the use of linear latent space models difficult and the direct application of kernel methods infeasible.

### 4.2  Experimental Setup

For path classification we employ a dictionary whose codewords are Histogram of Gradient Deviations (HGD) descriptors, as in [11]. This is well suited for characterizing tubular structures and can be applied in the same way to 2D and 3D images. This allows us, in theory, to apply a classifier trained on 2D images to 3D volumes. However, differences in appearance and geometry of the structures may potentially adversely affect classifier accuracy when 2D-trained ones are applied to 3D stacks, which motivates domain adaptation. We use half of the target domain for training and half for testing. 2500 positive and negative samples are extracted from each image through random sampling, as in [11]. This results in balanced sets of 30k samples for training in the source domain, and 20k for training and 20k for testing in the target domain.

To simulate the lack of training data, we randomly pick an equal number of positive and negative samples for training from the target domain. The HGD codewords are extracted from the road images and used for both domains to generate consistent feature vectors. We employ gradient boosted trees, which in our experiments outperformed boosted stumps and kernel SVMs. For all

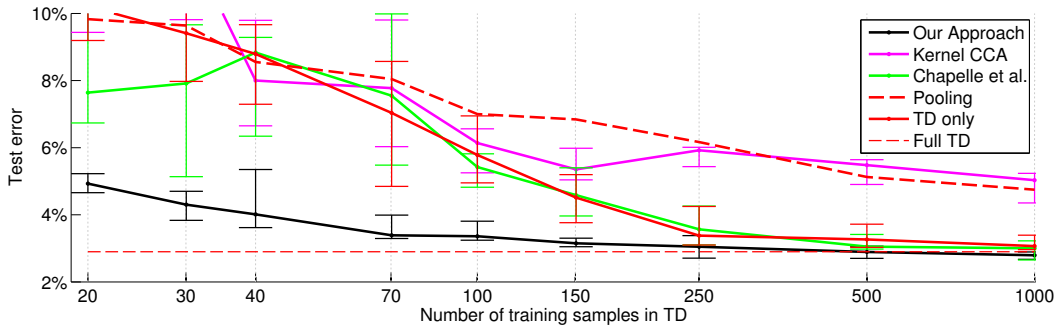

Figure 3: *Path Classification:* Median, lower and upper quartiles of the test error as the number of training samples is varied. Our approach nears *Full TD* performance with as few as 70 training samples in the target domain and significantly outperforms the baseline methods. Best viewed in color.

the boosting-based baselines we set the maximum tree depth to $L = 3$, equivalent to a maximum of 8 leaves, and shrinkage $\gamma = 0.1$, as in [8]. The number of boosting iterations is set to $K = 500$. For this dataset we report the test error computed as the percentage of mis-classified examples.

For mitochondria segmentation we use the boosting-based method of [27], which is optimized for 3D stacks and whose source code is publicly available. This method is based on boosted stumps, which makes it very efficient at both train and test time. Similar to [27], we group voxels into supervoxels to reduce training and testing time, which yields 15k positive and 275k negative supervoxel samples in the source domain. In the target domain it renders 12k negative training samples. To simulate a real scenario, we create 10 different transfer learning problems using the samples from one mitochondria at a time as positives, which translates into approximately 300 positive training supervoxels each. We use the default parameters provided by the authors of [27] in their source code ($K = 2000$), and we evaluate segmentation performance with the Jaccard Index, as in [27].

### 4.3 Baselines

On each dataset, we compare our approach against the following baselines: training with reference or target domain data only (shown as *SD only* and *TD only*), training a single classifier with both target and source domain data (*Pooling*), and with the multi-task approach of [10] (shown as *Chapelle et al.*). We evaluate performance with varying amounts of supervision in the target domain, and also show the performance of a classifier trained with all the available labeled data, shown as *Full TD*, which represents fully supervised performance on this domain and is useful in gauging the relative performance improvement of each method.

We compare to linear Canonical Correlation Analysis (CCA) and Kernel CCA (KCCA) [4] for learning a shared latent space on the path classification dataset, and use a Radial Basis kernel function for KCCA, which is a commonly used kernel. Its bandwidth is set to the mean distance across the training observations. The data size and dimensionality of the mitochondria dataset is prohibitive for these methods, and instead we compare to Mean-Variance Normalization (MVN) and Histogram Matching (HM) that are common normalizations one might apply to compensate for acquisition artifacts. MVN normalizes each input 3D intensity patch to have a unit variance and zero-mean, useful for compensating for linear brightness and contrast changes in the image. HM applies a non-linear transformation and normalizes the intensity values of one data volume such that the histogram of its intensities matches the other.

### 4.4 Results: Path Classification

The results of applying our method and the baselines for path classification are shown in Fig. 3. Our approach outperforms the baselines, and the difference in performance is particularly accentuated in the case of very few training samples. The next best competitor is the multi-task method of [10], although it exhibits a much higher variance than our approach and performs poorly when only provided a few labeled target examples. This is also the case for KCCA. The results of linear CCA are not shown in the plots because it yielded very low performance compared to the other baselines,

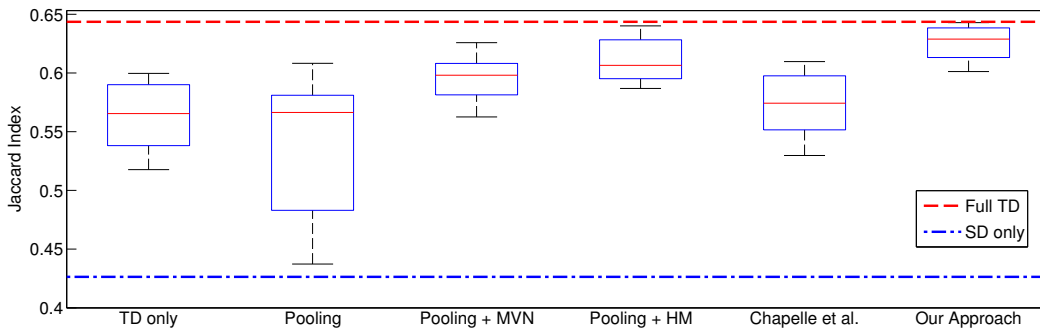

Figure 4: *Mitochondria Segmentation:* Box plot of the Jaccard index measure for our method and the baselines over 10 runs on the target domain. Simple Mean-Variance Normalization (MVN) and Histogram Matching (HM) although helpful are unable to fully correct for differences between acquisitions. In contrast, our method yields a higher performance without the need for such priors and is able to faithfully leverage the source domain data to learn from relatively few examples in the target domain, outperforming the baseline methods.

achieving a 14% error rate with 1k labeled examples and its performance significantly degrading with fewer training samples. Similarly, SD only performance is 16%.

Our approach is very close to *Full TD* in performance when using as few as 70 training samples, even though the *Full TD* classifier was trained with 20k samples from the target domain. This highlights the ability of our method to effectively leverage the large amounts of source-domain data. As shown in Fig. 3, there is a clear tendency for all methods to converge at the value of *Full TD*, although our approach does so significantly faster. The low performance of Chapelle et al. [10] suggests that modeling the domain shift using shared and task-specific boundaries, as is commonly done in multi-task learning methods, is not a good model for domain adaptation problems such as the ones shown in Fig. 1. This gets accentuated by the parameter tuning required by [10], done through cross-validation, that performs poorly when only afforded a few labeled samples in the target domain and yields a longer training time. The method of [10] took 25 minutes to train, while our approach only took between 2 and 15 minutes, depending on the amount of labeled target data.

## 4.5   Results: Mitochondria Segmentation

A box plot showing the distribution of the VOC scores throughout 10 different runs is shown in Fig. 4. Our approach significantly outperforms the multi-task method of [10] and the other baselines, followed in performance by pooling with mean-variance normalization (MVN) and histogram matching (HM). In contrast, our method yields higher performance and smaller variance over the different runs without the need for such priors. From a practical point of view, our approach does not require parameter tuning and cross-validation is not necessary. This can be a bottleneck in some scenarios where large volumes of data are used for training. For this task, training our method took less than an hour per run, while [10] took over 7 hours due to cross-validation.

## 5   Conclusion

In this paper we presented an approach for performing non-linear domain adaptation with boosting. Our method learns a task-independent decision boundary in a common feature space, obtained via a non-linear mapping of the features in each task. This contrasts recent approaches that learn task-specific boundaries and is better suited for problems in domain adaptation where each task is of the same decision problem, but whose features have undergone an unknown transformation. In this setting, we illustrated how the boosting-trick can be used to define task-specific feature mappings and effectively model non-linearity, offering distinct advantages over kernel-based approaches both in accuracy and efficiency. We evaluated our approach on two challenging bio-medical datasets where it achieved a significant gain over using labeled data from either domain alone and outperformed recent multi-task learning methods.

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
