[Reviews · NeurIPS 2013]

Submitted by Assigned_Reviewer_5

This paper proposed a new domain adaptation method based on boosting. Unlike most of the existing methods, the proposed method learns a single linear boundary in a common feature space and task-specific mappings from the original feature space to a higher-dimension common feature space
at the same time using boosting-trick.

[Quality]
The quality of this paper is high in terms of the proposed method and its effectivenss.

[Clearity]
The paper is written clearly. The organization is well.

[Originality]
The basic idea is novel, and the method is also well-designed. Instead of kernel-trick, it is interesting and very effectivethat boosting-trick is used.
Overall, the originality is high.

In the experiments, the proposed method showed the effectiveness for the settings where the source domain is 2D image data and the target domain is 3D image data,
although the existing domain-shift-based method does not work well for the same setting.

The method is parameter-free, which means it requires no parameter tuning. This is an advantege of the proposed methods.

Summary: The idea on learning a common feature space and mappings and its solution are highly original,
and its effectivenss was shown by the two experiments.
Therefore, this paper should be definitely accepted.

Submitted by Assigned_Reviewer_6

This paper addresses the problem of domain adaptation in bio-medical data based on boosting. The basic idea of the algorithm is to learn a non-linear transformation of different domains to a shared feature space where a single boundary shared across all tasks can be used for classification. The algorithm exploits the boosting trick to handle the non-linearities. The algorithm is applied on two challenging bio-medical datasets, achieving significant performance improvements in both.

Overall, I thought that the paper was well written, and the algorithm was novel enough to encourage further research in this direction. While the experiments show good results, I think the experimental evaluation was limited in the set of datasets used, and could be further improved by applying to a larger variety of datasets. Also, some potentially missing baselines and comparisons as mentioned below.

Specifically:
- It would be good to test the same algorithm on visual datasets such as the one by Saenko et al. Do the same findings hold in that case?
- Why isn't the method by Gopalan et al on unsupervised domain adaptation included as another baseline?
- What is the running time of your algorithm, for both training and testing?
- It would be good to compare your work against the following set of papers (mentioned with Z to be different from paper citations):
[Z1] Regularized Multi-Task Learning, KDD 2004
[Z2] Unbiased Look at Dataset Bias, CVPR 2011
[Z3] Undoing the Damage of Dataset Bias, ECCV 2012
Z1 is another popular work on multi-task learning that has a common and shared part in a max-margin framework. In addition, the problem seems to be closely related to Z2 and Z3. Specifically, in Z3, the authors also modify multi-task learning formulation for dataset bias (similar to domain adaptation) to learn a single classifier for all datasets.
Summary: Overall, I thought that the paper was well written, and the algorithm was novel enough to encourage further research in this direction. While the experiments show good results, I think the experimental evaluation was limited in the set of datasets used, and could be further improved by applying to a larger variety of datasets.

Submitted by Assigned_Reviewer_7

This paper presents an approach for multi-task learning. Existing approaches have focused on learning different decision boundaries for each task, where the boundaries are defined by a mix of shared and task-specific parameters. In contrast, this work proposes to use boosting to learn task-dependent non-linear feature mappings from each task to a common feature space, where a single linear decision boundary is learned and used for all tasks. Experiments demonstrate that the proposed approach outperforms alternative approaches on two tasks: path classification and mitochondria segmentation.

The paper is well-written, with the proposed approach, related work, and experimental evaluation all clearly explained.

The proposed approach appears to be a novel and technically sound method of using boosting to perform multi-task learning. In particular, the comparison against Chapelle et al. demonstrates the value of viewing multi-task learning as learning separate feature transformations from each task to a common space, as opposed to learning task-specific decision boundaries with some shared parameters.

Overall, the paper is well-written, the proposed approach is technically sound and interesting, and experiments empirically demonstrate the advantage of the proposed approach on two real-world applications. I believe this is a solid NIPS paper.

Minor comments:
- Provide definition and/or citation for Jaccard Index
Summary: Overall, the paper is well-written and present a novel formulation of multi-task learning. Experiments empirically demonstrate the advantage of the proposed approach on two real-world applications. I believe this is a solid NIPS paper.
Author Feedback

Author rebuttal: We thank the reviewers for their comments and address their remarks below.

R6
-----

- Datasets:
The datasets presented in the paper are both challenging and representative of classification problems that involve very large 3D data volumes, such as those required by bio-medical applications where obtaining labeled data is very costly. However, we agree that further evaluation on additional datasets, such as those of Saenko et al. is worthwhile, which we plan to do as part of future work.

- Train and test times:
As stated in section 4.5 (Mitochondria Segmentation), our approach takes less than an hour to train, which is much faster than the baseline of [6], due to the cross-validation step needed by the latter. Similar observations hold for the path dataset, which we will include in the final version of the paper if accepted. Furthermore, the test time for our approach is similar to the test time of any of the other boosting baselines used, since test-time complexity depends on the structure of the boosted classifier, which is kept the same for each dataset (i.e., the number of iterations and weak learner type).

- Comparison with [Z1] and [Z3]:
[Z1] and [Z3] extend the binary SVM classifier to the multi-task case, with the assumption that tasks are related through shared and task-specific weight vectors (see Fig 2(a) in our paper). Note that [Z1] and [3] are the conference and journal versions of the same approach, respectively. In our paper we evaluate our approach against a boosting-based formulation of [3], namely Chapelle et al. [6]. In contrast to [3] that uses the kernel-trick, the boosting-trick of [6] makes it feasible to apply it to large datasets with high dimensional feature spaces, which is a primary focus of our paper. Moreover, the methods of [6] and [3] performed similarly when evaluated in [6].

- Comparison with [Z2]:
[Z2] discusses the issue of dataset bias in computer vision and possible guidelines on how to acquire more representative datasets. However, in our setting we have fairly little control over how the data is acquired. The focus then becomes less on how to improve the data collection process, which is the topic of [Z2], and more on how to develop algorithms that can effectively leverage the training data despite its inherent bias. Also, an important issue not addressed by [Z2,Z3] is the cost of obtaining labeled data. In our problem obtaining labeled data is very costly and is typically scarcely available in one or more domains, which introduces additional challenges not considered by [Z2,Z3].

- Comparison with Gopalan et al.:
As in our approach, Gopalan et al. introduces a shared latent space to model the transformation between domains. However, their method is linear and therefore less suited to problems involving nonlinear transformations, such as those considered in our work. In our experiments, this is evidenced by the fact that linear CCA far underperformed the other non-linear methods including K-CCA.

R7
----

We will add a citation for the Jaccard Index.